

# Analysis of lightning outliers in the EUCLID network

Dieter R. Poelman[1], Wolfgang Schulz[2], Rudolf Kaltenboeck[3], Laurent Delobbe[1]

[1]Royal Meteorological Institute of Belgium, Brussels, Belgium
[2]OVE-ALDIS, Vienna, Austria
[3]Austro Control, Innsbruck, Austria

*Correspondence to*: Dieter R. Poelman (dieter.poelman@meteo.be)

**Abstract.** Lightning data as observed by the European Cooperation for Lightning Detection network EUCLID are used in combination with radar data to retrieve the temporal and spatial behaviour of lightning outliers, i.e. discharges located on a wrong place, over a 5-year period from 2011 to 2016. Cloud-to-ground stroke and intracloud pulse data are superimposed on corresponding 5-min radar precipitation fields in two topographically different areas, being Belgium and Austria, in order to extract lightning outliers based on the distance between each lightning event and the nearest precipitation. It is shown that the percentage of outliers is sensitive to changes in the network and to the location algorithm itself. The total percentage of outliers for both regions varies over the years between 0.8% and 1.7% for a distance to the nearest precipitation of 2 km, with an average of approximately 1.2% in Belgium and Austria. Outside the European summer thunderstorm season the percentage of outliers tends to increase somewhat and could result from the fact that more sensor upgrades occur during winter or that precipitation of winter thunderstorms is underestimated compared to the vertically extended summer storms. The majority of all the outliers are low peak current events with absolute values falling between 0 to 10 kA. More specifically, positive cloud-to-ground strokes are more likely to be classified as outliers compared to all other type of discharges. Furthermore, it turns out that the number of sensors participating in locating a lightning discharge is different for outliers versus correctly located events, with outliers having the least amount of sensors participating. In addition, it is shown that in most cases the semi-major axis assigned to a lightning discharge as a confidence indicator in the location accuracy is smaller for correctly located events compared to the semi-major axis of outliers.



# 1 Introduction

Present-day lightning location systems (LLS) are the result of continuous development over the years with improved location accuracy, peak current estimation and type classification for each observed lightning event. However, despite the great progress made to determine those properties amongst others, occasionally some events remain poorly determined by the LLS. For instance, the uncertainty of the measurements related to a low peak current discharge tends to be larger than it is for a high peak current event. In addition, it is still common practice to categorize positive cloud-to-ground (CG) strokes with estimated peak currents smaller than 5 or 10 kA as IC pulses since those are more likely to be of intracloud (IC) nature (Cummins et al., 1998; Wacker and Orville, 1999a, b; Jerauld et al., 2005, Orville et al., 2002; Cummins et al., 2006; Biagi et al., 2007). However, not all the properties are of equal importance for the different users of lightning data. Depending on the customers' application of the LLS data, different performance features are more, while others are less important, e.g. power utilities normally do not care about the IC detection efficiency (DE) of a LLS, whereas the quality of the CG data is of utmost importance. On the other hand, aviation control and meteorological services which often trigger warning messages based on LLS data favor a good DE of CG as well as IC events coupled to a minimum of events located on a completely wrong position. It is therefore a necessity to gain a thorough knowledge of the LLS at hand.

During recent years the performance of LLS got more and more attention (Nag et al., 2015). A direct method to determine the quality of a network, and therefore the values assigned to each lightning event, is by comparing the data against so-called ground-truth observations. Those observations provide valuable information on the DE, location accuracy (LA) and in some cases even the peak current estimates retrieved from an LLS. This is done for instance by examining direct lightning strikes to instrumented towers (Diendorfer et al., 2000a, 2000b; Pavanello et al., 2009; Romero et al., 2011; Schulz et al., 2012; Schulz et al., 2013; Cramer and Cummins, 2014; Azadifar et al., 2016), through the use of rocket triggered lightning (Jerauld et al., 2005; Nag et al., 2011; Chen et al., 2012; Mallick et al., 2014a, 2014b, 2014c), and/or by recording lightning strikes with high-speed video and E-field measurements in open field (Biagi et al., 2007; Chen et al., 2012; Poelman et al., 2013a, Schulz et al., 2016). Although best practice to retrieve robust information on a networks' performance, the



aforementioned methods are quite labor intensive in order to acquire a large enough dataset for a statistically reliable output. Other methods exist, such as intercomparing different LLS within regions of overlapping coverage (Said et al., 2010; Pohjola and Mäkelä, 2013; Poelman et al., 2013b). However, the main disadvantage of those studies is the assumption of one network as being "ground-truth". In reality this is hardly the case for any existing LLS, except maybe for the short-baseline lightning mapping arrays (Rison et al., 1999; Thomas et al., 2004; van der Velde et al., 2013; Defer et al., 2015).

In this paper lightning data are combined with radar precipitation observations to analyze the temporal and spatial behavior of lightning outliers in two topographically different regions in Europe. Lightning outliers are sometimes also referred to in the literature as fake or ghost strokes and can be the result of signal interferences from power lines, radio frequencies or other site-specific disturbances or are simply misplaced events by the location algorithm. The results presented here are obtained by combining lightning observations from the European Cooperation for Lightning Detection network EUCLID with radar precipitation data in Belgium and Austria, as described in Section 2. The results of the analysis are presented in Section 3 and summarize in Section 4.

## 2 Data and Methodology

### 2.1 Lightning location data

The European Cooperation for Lightning Detection network EUCLID has been operational since 2001 and processes as of January 2017 in real-time data of 164 sensors to provide European wide lightning observations of high and nearly homogeneous quality (Poelman et al., 2016). All of the sensors operate over the same low-frequency (LF) range and provide amongst others timing and angle information. The individual raw sensor data are sent in real-time to a single processor, calculating the electrical activity at any given moment. The locations of the EUCLID sensors are displayed in Fig. 1. The network has been tested continuously over the years against ground-truth data from direct lightning current measurements at the Gaisberg Tower in Austria (Schulz et al., 2016), Peisserberg Tower in Germany (Heidler and Schulz, 2016) and Säntis Tower in Switzerland (Romero et al., 2011; Azadifar et al., 2016) and data from E-field and video recordings in Austria, France and Belgium (Schulz et al., 2016). The latest





comprehensive performance analysis of the EUCLID network based on those measurements revealed that the flash and stroke DE for negative CG discharges in different regions of the EUCLID network are greater than 93% and 84%, respectively, while for positive events those are greater than 87% and 84 %, respectively (Schulz et al., 2016). To retrieve the latter values, only those strokes are used in the

analysis that match certain quality criteria such as $\chi 2$ and semi-major axis of the confidence ellipse, and received a correct stroke classification as CG by the central processor. Those strict criteria, as well as temporary sensor outages during the measurements campaign, can impact the DE estimates given in Schulz et al. (2016). In addition, Schulz et al. (2016) showed that the LA dropped steadily over the years down to the present LA in the range of 100 m within the majority of the network. Note that in

Schulz et al. (2016) ground truth observations are collected in Austria and Belgium, the same regions of the EUCLID network which are under consideration in this paper.

During the time period under consideration, significant changes of the EUCLID network regarding DE and LA were made (Schulz et al., 2016). Those are related to new sensor technology, timing error corrections and a new location algorithm which can influence the outlier behavior. One would think

sensor upgrades have always a positive influence on a networks performance. While this is generally true in the long run, the upgrades can cause temporary problems in the beginning since those sensors are awaiting calibration. This is especially true for some sensors in Italy in 2014. From the day of the setup till the sensors were calibrated those sensors were configured to provide timing information only. Timing only sensors often increase the number of outliers.


## 2.2 Weather radar data

Weather radar data of the Royal Meteorological Institute of Belgium (RMIB) and of Austro Control in Austria are used in this study. Fig. 1 shows the locations (white stars) and coverage (dashed lines) of the individual radars, as well as the limit of the composite as the outer contour of all the radars (solid lines).

The use of radar composites is preferred over the individual radar observations since individual radar observations can be hampered by shielding effects. This is true especially in mountainous regions such as the Alps in Austria, limiting the detection range where the radar data is still considered of sufficient




quality. Moreover, further away from the radar the precipitation can be underestimated due to overshooting beams. The composite radar reflectivity threshold is set at 12 dBZ. Following the

$Z=200*R^{1.6}$ relationship from Marshall and Palmer (1948), with Z being the reflectivity and R the rain rate, this threshold corresponds to a rain rate 0.2 mm/h below which the rain rates are set to zero in this study. This low reflectivity threshold helps to detect convective clouds relevant for lightning generation even in weak cell cores from winter events or upper areas of thunderstorms at far ranges from the radar site.


### 2.2.1 Belgium

The radar composite used at RMIB consists out of three radars. RMIB owns and operates two of them; the radar at Wideumont in the southeast of Belgium and the radar in Jabbeke located near the west coast which became only operational since 2013. The third weather radar at the center of the composite is

located at the airport in Zaventem near Brussels and is operated by Belgocontrol, in charge of the safety of civil aviation. All of the radars are C-band Doppler radars, performing a multiple elevation reflectivity scan every 5 minutes with a resolution of one degree in azimuth and 500 m in range for Jabbeke and Zaventem and 250 m in range for Wideumont. The maximum range is 300 km for Jabbeke and 250 km for Zaventem and Wideumont. A Doppler filter for clutter elimination is used for the three

radars and an additional polarimetric fuzzy logic filter is used for Jabbeke. For each radar a 2D precipitation product is derived from the volume reflectivity data. A composite is subsequently produced from these 2D products taking for each pixel the maximum value of the radars covering this pixel.

### 2.2.2 Austria

Austro Control, the Austrian civil air service provider is operating five C-band EEC polarized Doppler weather radars in Austria, of which four of them are used in this study. Two of the radar sites are located on the foothills of the Alps close to Vienna and Salzburg (Rauchenwart and Feldkirchen) and the other two radar sites are situated in the west and south of Austria at mountain tops above 2000 m

close to Innsbruck and Klagenfurt (Patscherkofel and Zirbitzkogel). The underlying volume scan





contains 16 elevations ranging between -1.5° and 67° up to a range of 224 km. Doppler and statistical clutter filters are applied before creating maximum surface projection of reflectivity which combines strongest return from each elevation level. Resulting Austrian composite uses maximum reflectivity in horizontal extent which is provided by one of the 4 radars to avoid shielding effects of the Alps.

Temporal and spatial resolution is 5 minutes and 1 km, respectively.  For more details, the interesting reader is referred to Kaltenboeck and Steinheimer (2015) and Kaltenboeck (2012a, 2012b). It is important to note that the Austrian weather radar network was upgraded between 2011 and 2013, during which the individual radar gains were modified. This adaptation of the gain could easily influence to some degree the findings in this paper.


## 2.3 Methodology

To account for border effects, only lightning events within the red boxes as indicated in Fig. 1 are used. Those regions correspond approximately to the area where two or more radars participate in the radar image with sufficient distance from the border. Subsequently, CG strokes and IC pulses are

superimposed on corresponding 5-min radar precipitation fields. In order to have overall homogeneous coverage of the weather radar data, only the time steps were used for which all the radars within the composites were in operation. An event is then categorized as an outlier when no precipitation within a certain distance has been observed. The distance at which an event is classified as outlier is somewhat chosen arbitrary. Different runs are performed applying a distance $\Delta r$ of 2, 5, and 10 km. An example of

this method is visualized in Fig. 2. All the lightning events are superimposed as black dots with the retrieved outliers in red for clarity. This method is supposed to give a lower limit of the percentage of outliers because some of the outliers will, by chance, be placed in a region with radar reflectivity larger than 12 dBZ. In the remainder of the paper, explanation of the results is based on the findings for a search radius $\Delta r$ of 2 km, unless otherwise stated explicitly.




## 3 Results

The overall annual percentage of outliers for CG strokes and IC pulses relative to the total number of events, as a function of Δr between the event location and the nearest precipitation, is plotted in Fig.3 for Belgium and Austria, respectively. There are several similarities and differences between the two

areas. For example, the total percentage of outliers is of the same order of magnitude for both regions and varies between 0.8% and 1.7% throughout the years for an adopted Δr of 2 km. It is clear that choosing a larger Δr decreases the percentage of outliers, and vice versa, while maintaining the same annual trend. The percentage of the total outliers averaged over six years in Belgium and Austria is approximately 1.2%. In Belgium on average 0.5% of the outliers are of type CG and this value increases

up to 0.9% in case of IC, whereas in Austria the level of CG outliers is only slightly higher than that of IC outliers, i.e., 0.8% for CG with respect to 0.5% in case of IC. Shorter baselines in Austria compared to Belgium could be a reason for this discrepancy. The significant higher number of outliers in Belgium in 2011 compared to Austria can be attributed to a timing only sensor located close to Belgium (Den Haag) and another sensor in the Netherlands which was moved and afterwards operated for a longer

time period with deactivated angle information (Roermond). From our experience, sensors providing only time information often cause additional outliers. For the vast majority of the sensors which provide angle and time information those measurements have to be consistent since coherence between the latter two reduces the number of outliers. The level of outliers from 2012-2014 is roughly the same for both areas. The lowest level of CG outliers is found in 2016 in both areas. In addition, it is worth mentioning

that in Belgium and Austria the majority of the CG outliers are single stroke flashes, while only a minority of the CG outliers belong to a flash with multiplicity larger than one. One could say that assuming a stable radar network, the variation in the percentage of outliers over the years reflects the status of the lightning location network in a certain area.  Hence, continuous monitoring of the outliers has the possibility to pick up potential problems in the network, which can be relevant for future

automatic forecast applications.

Figure 4 reveals the spatial distribution of the percentage of total outliers observed in between 2011 and 2016 on a 10 x 10 km$^2$ grid. In Belgium, values range from 0.2% to 3.3%. The distribution of outliers within Belgium is rather uniform and lies between 1% and 2%, with here and there somewhat higher



percentage values. The latter are mainly caused by IC outliers, since those contribute the most to the

overall outlier percentage in each grid cell. In Austria, grid cell percentages range from 0.1% up to 33%. The majority of the grid cells have low outlier percentage values, except in the southwest corner. This is exactly the place where the Alps disturb the radar observations leading to an increase in outliers with the employed method.

Fig. 5 illustrates the monthly variation of the percentage of outliers, in addition to the absolute number

of observed lightning events. An obvious decrease is observed in the percentage of outliers during May-Sept, compared to the other months of the year. This feature could be either related to the fact that more sensor upgrades occur during winter or because precipitation of winter thunderstorms is more difficult to detect with the weather radars. Regarding the sensor upgrades, those often result in disabled angle information because systematic angle errors, i.e. site errors, are at first unknown and the correction

takes a while because lightning data is necessary. Consequently, upgraded sensors start operation with disabled angle information during winter months. With respect to the observation of precipitation, during summer most of the storms are associated with large amounts of precipitation in vertically extended clouds, meaning that those storms are always very well detected by the radars. In contrast, winter storms are generally associated with less intense precipitation cells and with smaller vertical

extensions. In some cases such storms will be hardly detected by the radars at long range. Nevertheless, wrong classification of lightning as outlier due to the non-detection by the radar is probably extremely limited. A wrong classification may also occur when a wrong detection appears by chance in a precipitation area detected by the radar. Since radars generally detect less precipitation in winter than in summer (e.g. Hazenberg et al., 2011) such wrong classification occurs less in winter than in summer

wich means that the classification method will produce more outliers in winter. Note that Poelman et al. (2016) showed that on average peak current estimates of winter lightning are higher than in summer. One would therefore expect that on average in winter more sensors participate in a lightning event compared to summer, resulting in a good location accuracy. Nevertheless, the absolute number of outliers during winter is much smaller compared to summer, as can be deduced from the grey-scaled

monthly distribution of total observed events in Fig. 5. Thus, the increase in percentage of outliers may not be too important for the majority of applications.





Fig. 6 plots the outlier percentages related to each individual group, e.g. percentage of negative IC is related to the total number of negative IC. Proportionally, the degree of occurrence of positive and negative outliers is of the same level, except for 2011, and follows the annual variation as in Fig. 3.

Positive CG strokes exhibit the highest percentage of outliers in Belgium and Austria. This could be related to the fact that positive CG strokes are often accompanied with significant IC activity complicating the transmitted electromagnetic fields (Fuquay, 1982; Saba et al., 2009). It is therefore harder to detect and locate correctly such strokes, resulting in a higher percentage of outliers. Furthermore, the percentage of negative CG outliers is roughly half of that of the positive CG outliers

for the years 2011-2014. The opposite is found in case of IC pulses, where the percentage of negative IC outliers is higher compared to the positive counterpart. However, the difference between positive and negative CG outliers and/or IC pulses decreases in 2015 and 2016. Thus, the percentage of outliers is more or less unrelated to the polarity of the event. In 2016, it is obvious that the outlier percentages of the individual types are more or less in line with each other. This could also be a result of the improved

performance of the latest adopted location algorithm.

In Fig. 7, the percentage of outliers for peak current intervals up to +/- 20 kA is plotted, calculated with respect to the total amount of discharges within each peak current interval for all the 6 years of data. It is seen that the distribution corresponds well between Belgium and Austria. Because positive CG strokes with peak currents below 5 kA are categorized as IC, no data for positive CG below 5 kA exists.

First of all, the majority of the outliers for positive CG and IC discharges are found in between [5, 10] kA and [0, 5] kA, respectively, with a decline towards the larger peak current intervals. This is not surprising since the higher the peak current, the more sensors participate on average in locating the event. This is also true for negative IC outliers, whereas negative CG outliers have the highest percentage in the [-10, -5] kA range. Except for the [-5, 0] kA interval, the percentages are similar

between negative IC and CG outliers. This is not the case for the positive outliers.

Fig. 8 reveals the cumulative distribution of the number of sensors participating in a solution as a function of event type. First of all, one notices that in case of CG strokes more sensors participate in a solution compared to IC events. This is attributed to the fact that in the LF range the amplitude of even the largest IC pulses is significantly lower compared to that of CG return strokes (Weidman et al.,

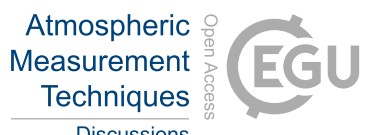

1981). The amplitude difference between CG strokes and IC pulses increases even further with increasing propagation distance between the source and the lightning sensor (Cooray et al., 2000). Hence, more sensors will detect the radiation from a single CG discharge compared to an IC pulse. The resemblance in distribution between Belgium and Austria is not surprising since the lightning sensors in EUCLID are quite homogeneously distributed across the network. In addition, more sensors participate

in the location of discharges that are correctly located, than is the case of outlier CG as well as IC events. For instance, 85% of the IC outliers are located by 2 or 3 sensors, whereas this drops to 50% for correctly located IC pulses. For CG strokes on the other hand, only 20% of the outliers are located with more than 6 participating sensors, whereas this is the case for more than 60% for the CG strokes within 2km of the nearest precipitation. We find that the median amount of sensors participating in a solution

for correctly located CG and IC is 8 and 3, respectively and this drops to 3 and 2 sensors participating in case of CG and IC outliers.

Finally, the central processor assigns to each lightning event a value of the semi-major axis (SMA) of the 50% confidence ellipse. This value can be used as a quality indicator of the location accuracy, with smaller values indicating a larger confidence in the assigned location of the event. The distribution of

SMA for all the events (CG + IC) is plotted in Fig. 9, separated into outlier and correctly located events. Note that events with an SMA larger than 7.5 km do not exist in the data since those events are rejected by the location algorithm because those are seen as bad quality events. First of all, it is striking that the SMA distribution is almost equal for Belgium and Austria. The majority of OK events, i.e. 75%, have SMA values falling in the 0-1 km range, whereas this drops to 40% in case of outliers. The average and

median value of the SMA for OK events is 775 m and 200 m, respectively, and this increases to 1.83 km and 1.48 km in case of outliers. Although not shown in this plot, it is found that the average SMA for CG strokes is smaller by a factor of two compared to IC pulses. This is expected since more sensors participate in a solution for CG strokes compared to IC pulses as discussed in Fig. 8. A similar approach was used to find a correlation between the $\chi 2$ value; a measure for the correspondence between the

different sensor measurements, although with no clear result. Hence, this prevents us from using SMA and $\chi 2$ parameters to help in outlier classification.





Looking at Fig. 7 to 9 one could wonder whether those CG outliers could be simply considered as IC discharges misclassified by the network, since IC discharges have on average lower peak currents, hence lower number of contributing sensors and therefore smaller SMA. Although this can be partly
true, still a considerable fraction of the CG outliers are found to have large peak currents. It is therefore unlikely that all the CG outliers found with this method are in fact misclassified IC discharges.

## 4 Summary

In this study all lightning events detected by the EUCLID network during 2011 and 2016 that fall within selected areas in and around Belgium and Austria are classified as outliers or correctly located events
based on their distance $\Delta r$ to the nearest precipitation. Categorizing the events with the aforementioned technique based on weather radar data and comparing the results from different geographical regions is not a straightforward task. The reason is potential calibration issues in the different radar networks with maybe even different technology and local beam blockage problems especially in the mountainous regions in Austria. A workaround, at least for the last problem, is to use composite radar data.
Independent of those difficulties the overall results in both regions agree quite well. The overall percentage of outliers for both regions is between 0.8% and 1.7% in the individual years for a distance $\Delta r$ to the nearest precipitation of 2 km and drops further when a more relaxed $\Delta r$ is chosen. These values are lower limits since it is possible that an outlier is located in an area with rain. The percentage of outliers is quite small having in mind that a $\Delta r$ of 2 km is already a quite strict criteria. Outside the
European summer thunderstorm season the percentage of outliers tends to increase somewhat. This increase could be related to the fact that the radar underestimates or might not detect to some extent precipitation. This in turn is related to the fact that in general winter storms are less vertically developed compared to summer storms and show lower cell core reflectivity. The majority of all the outliers are low peak current events with absolute values falling between 0 to 10 kA. More specifically, positive CG
strokes are more likely to be classified as outliers compared to all other type of discharges. Furthermore, it turns out that the number of sensors participating in locating a lightning discharge is different for outliers versus correctly located events, with outliers having the least amount of sensors participating. In




addition, it is found that in general the SMA of non-outliers is much smaller compared to the SMA belonging to outliers.

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









**Figure 1: The locations of the EUCLID sensors in the domain are indicated (black dots), as well as the positions of the radars (white stars) together with the respective collective detection range in Belgium and Austria. The red boxes indicate the two areas that are used in this study.**







Figure 2: Example of a 5-min precipitation field, superimposed with the lightning events within the time interval. The 'true' events are indicated as black dots, whereas the derived 'outlier' is plotted in red. For clarity the underlying precipitation field has been given the same value above the applied threshold of 0.2 mm/h everywhere.







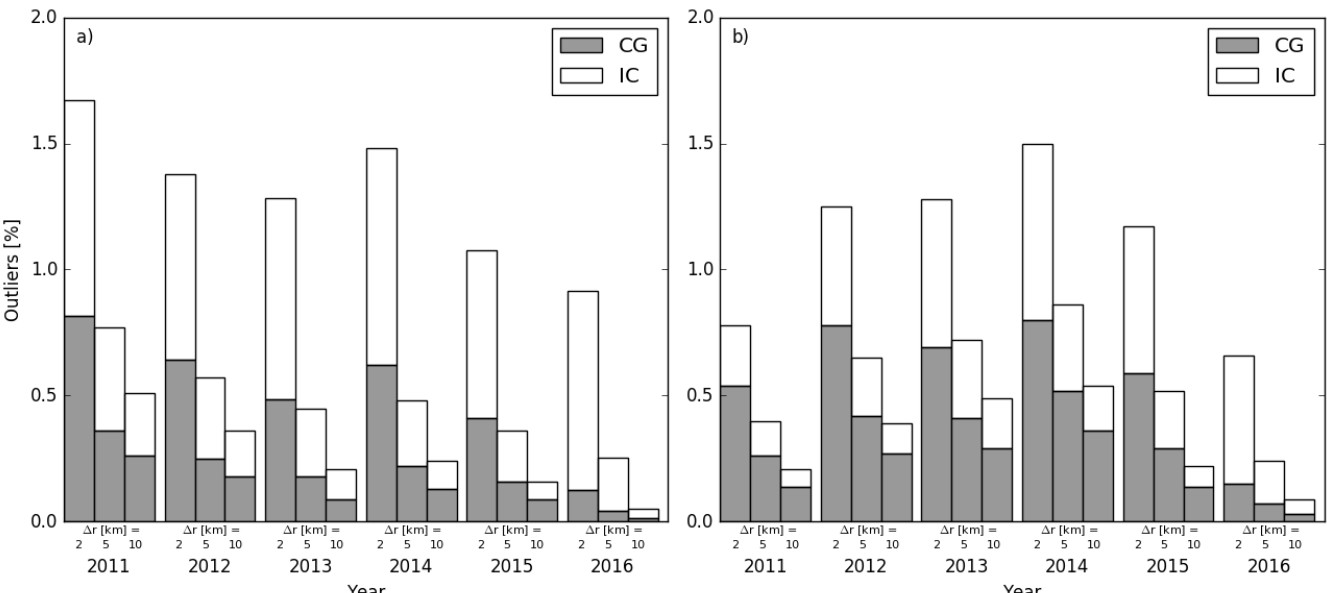

**Figure 3: Annual variation of outliers in a) Belgium and b) Austria, based on cloud-to-ground (CG) and intracloud**
**(IC) events, for search radii of 2, 5, and 10 km, respectively.**







**Figure 4: Spatial distribution of the total percentage of outliers in Belgium (top) and Austria (bottom) for an adopted search radius of 2 km on a 10 x 10 km$^2$ grid.**





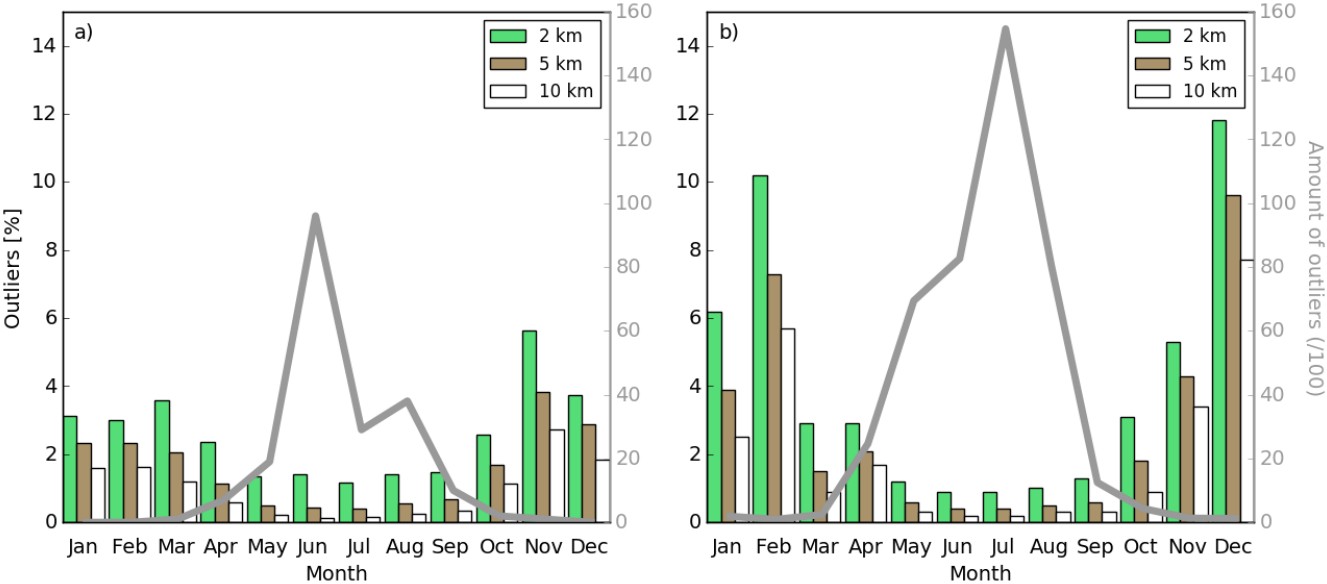


**Figure 5: Monthly distribution of the total (CG + IC) percentage of outliers in a) Belgium and b) Austria, for search radii of 2, 5, and 10 km, respectively. In addition, the total number of lightning detections is plotted as well (grey scale).**








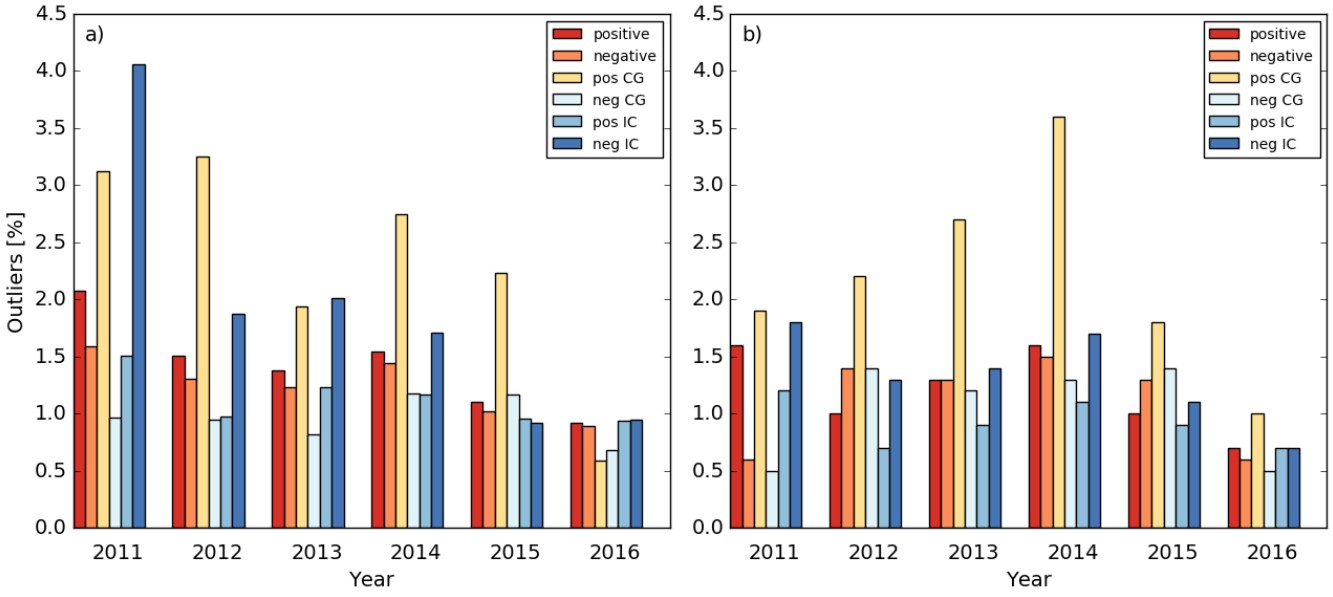


**Figure 6: Percentage of outliers versus event type in a) Belgium and b) Austria, for a search radius of 2 km.**






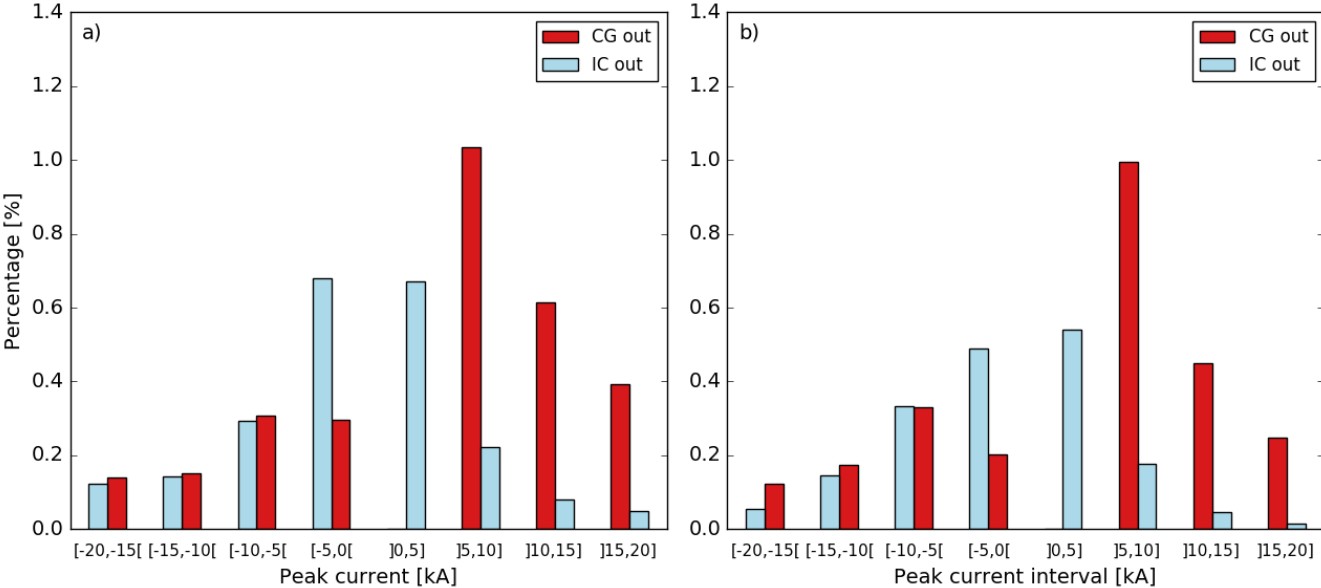

**Figure 7: Percentage of outliers as a function of peak current in a) Belgium and b) Austria, for a search radius of 2 km.**











**Figure 8: Cumulative distribution of the number of sensors participating in a solution for CG and IC outliers ('out'))**
**and correctly ('ok') located events (solid=Belgium, dashed=Austria).**


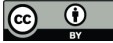




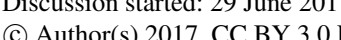

**Figure 9: Distribution of the semi-major axis (SMA) in Belgium and Austria, for a search radius of 2 km.**

