# Peer review of "Analysis of lightning outliers in the EUCLID network"

_Atmospheric Measurement Techniques, 2017_

## Referee Comment (RC1) · K. Naccarato (Referee) · 6 Jul 2017

It is an interesting analysis of lightning solutions provided by the EUCLID network that sometimes do not accurately match the precipitation patterns given by weather radar images. The manuscript is well written, figures are clear and well explained and discussions are comprehensible. Anyway, I have some comments on 3 specific points:

1) In line 84, I really do not understand the sentence: "Note that the latter values are impacted by the strict location quality criteria and correct required stroke classification, i.e. CG versus CG, used in the analysis, as well as temporary sensor outages during the measurements campaign". Please clarify.

2) From line 193 to 213, the authors discuss the results of Figure 5 which mainly

shows the seasonal variation of the percentage of outliers. According to the data, clearly during the winter time there is an increase in the number of outliers due to mainly 2 factors: (1) sensor upgrades that provides only TOA solutions during the calibration period; (2) low reflectivity of the precipitating systems due to their smaller size and depth. However, the discussion is confused and I cannot clear understand the apparently 2 opposite effects and whether they are important or not: (1) the higher percentage of outliers during winter and (2) the higher absolute number of outliers during summer. I suggest this discussion to be rewritten to improve clarity.

3) From Figures 7, 8 and 9, I ask to the authors: all those outliers cannot be considered simply IC discharges (misclassified or not) by the network? Note that they mostly present the typical behavior of IC flashes:(1) low peak current values (because they are in majority weaker than the CGs); (2) usually are detected with larger SMA because are detected by less sensors and has long horizontal extensions inside the clouds leading to major errors in their location, i.e., projection over ground; and (3) present (in a such way) "random" polarity since the ICs can move upward and downward inside the clouds (mostly upward we know!). I'd like to hear more from the authors about this point based on the presented results.
* * *

---

## Referee Comment (RC2) · Anonymous Referee #1 · 24 Jul 2017

The paper describes a method to distinguish lightning outliers from the typical EU-CLID cloud-to-ground and intra-cloud lightning dataset based on radar-based 5-min precipitation product. The methodology consists in merging the lightning data to the radar-based precipitation data and to study the properties of the lightning outliers located in a non precipitating region based on a spatial criterion. The properties are then discussed according to different information available from the lightning observations.

I think the methodology needs a bit more description, and more specifically on the way the advection is taken into account in the lightning dataset and on the beginning and ending times of the 5-min period used to categorize the different EUCLID lightning records.

The sensibility and accuracy of the radar-based precipitation products should also be

discussed as it is served as reference and limitation should be identified and taken into account during the analysis. I also wonder if rain field is the proper radar-based parameter to investigate. I would have also looked at the radar reflectivity composite as lightning flashes can propagate outside the rain field. Looking only the rain field suggests implicitly that you are considering that EUCLID records are predominantly located in the cores or closed to the convective cores. Using the radar reflectivity composite would help you to identify outliers in cloud-free regions to outliers in cloud regions.

What is also missing is some information on when during the flashes the different outliers are detected. Are they detected during the preliminary breakdown or latter during the life of the flashes? And to which processes do they correspond? I believe that such question can only be investigated by comparing EUCLID data to LMA-type observations. Note that a comparison with LMA-type observations might also require looking at the raw LMA data recorded by each station.

Note that the Authors do not exploit the value of rain accumulation that is provided by the radar-based product. I would have added some statistics on that parameter according to the properties of outliers and non-outliers.

Finally, no discussions are given by the Authors on how they could improve their algorithm based on the present analysis but maybe it is already planned or under way.

The manuscript is well written even if some clarifications are required. I think some additional parameters should be inserted in the figures. Please find below some specific comments.

Line 19. One should not forget that the 3D structure of the flashes might be different in winter compared to the one in summer.

Line 38. Is there a missing word after "more"?

Lines 64-67. Why only these two regions? And not a larger domain covered by both

EUCLID and the European radars?

Line 98. Please provide some physical and/or technological explanations on your statement that "timing only sensors often increase the number of outliers".

Line 108. What do you mean by "overshooting beams"? Please rephrase.

Lines 108-113. I suspect the precipitation product you have been using has been validated. It might be relevant to provide some references on such validation in your paper.

Line 109. Is the Marshall-Palmer relationship valid whatever the precipitation regime? I suspect in your case you are more interested in low precipitation amount where potentially you might find the lightning outliers. So do you think that the radar product used here is sensitive enough to deliver a reliable and accurate product for your investigation? Why did you choose a radar-based precipitation product and not for example the reflectivity composite? Discharges are not only propagating where precipitation occurs (e.g. spider lightning). So I wonder if your choice to use the radar-based precipitation product does not lead to a larger uncertainty. Do you have any comment?

Lines 116-127. How is the advection taken into account as a 5-min precipitation product is generated? And how do you take into account the advection in your lightning data? At which altitude does the radar-based precipitation product correspond?

Lines 130-143. Same questions as for lines 116-127. How is the advection taken into account? At which altitude does the radar-based precipitation product correspond? Is the precipitation product comparable in terms of accuracy for both domains of interest?

Line 146. How is the radar-based precipitation distributed in those two domains? Are they geographically uniformly distributed? In Figure 4, you are giving the spatial distribution of the % of outliers. I would have added with iso-contours (in white) the actual lightning distribution from where you computed the %.

Lines 148-149. Again how did you take into account the advection of the precipitation
and of the lightning activity? The radar product is provided per 5-min period. How did you select the lightning data within that 5 min period and how does it fit with the way the radar product is built?

Lines 151-152. Again this suggests that precipitation at the ground is required to verify the lightning data. How are we sure that precipitation is always required where lightning flashes occur? Please provide some arguments to strengthen your methodology. I think a similar analysis using the reflectivity composite should be performed in order to identify cloud-free outliers to in-cloud non-precipitation outliers. At which stage of the lightning flash do the lightning outliers correspond?

Line 153. I suspect you have projected the lightning data on the same temporal and spatial grid as the radar data, right?

Figure 3. Could you please plot as well the number of CG strokes and IC pulses in order to see how your dataset spreads over the years?

Lines 161-163. Am I correct when I say that you did not try to associate the outliers with "correct" events based on a time criteria for example? I wonder if a time criteria should not be added to your analysis in order to dissociate isolated events to outliers mis-located from a group of events.

Line 185. See comment on Line 146 for Figure 4.

Lines 194-196 Could not we say as well that the October-to-April flashes might have different vertical and horizontal structures that make them detected by EULCID sensors with more difficulties? One should not only criticize the radar sensibility or the sensor upgrade.

Lines 203- 204. I understood that the two geographical domains you have been studying should not really suffer of any long range issue. Am I right?

Line 208. "which" instead of "wich".

Lines 212-214. I agree with you on the climatology point of view, but then it depends on the application you want to do with your outliers data.

Lines 217-219. This confirms the interest of considering a temporal criterion in your analysis in order to discriminate isolated (in time and space) outliers to isolated in space only outliers. What do you think?

Figure 6. Without the actual number of events considered for each year, it is difficult to identify how statistically representative is the dataset you are studied in Figure 6. Could you please add that information?

Line 228. What the % of outliers with an absolute current above 20 kA?

Line 238. I would add "positive current outliers" instead of "positive outliers".

Lines 228-238. I have a question about all those low current events. How confident are you in their detection and on their classification not only in terms of IC or CG but also in terms of polarity?

Lines 238-253. How accurate are the event locations when two lightning sensors are only used? Do you usually keep flashes detected with only two EUCLID sensors? Have you plotted the same parameter but by range of current? I would like to see how the number of lightning sensors influences the detection of the outliers according to their estimated current. You could plot it with 2D cumulative distribution.

Line 257. Do you see any difference between IC and CG outliers separately?

Lines 256-259. Do you have a way to get the number of events that were rejected by the central processor for the period of data you have studied?

Figure 8. The number of samples per SMA range would provide an idea on the statically representativeness of the dataset used here. Please add that parameter in Figure 8.

Figure 8 (continued). Similarly to what I suggested for Lines 238-253, have you looked

on how the SMA is distributed according to the number of lightning sensors used to locate the different event categories? If yes, what are your main conclusions? If not, please take a look on that plot and provide some information in response to the present comment.

Lines 265-267. I do not understand your statement. Please explain it.

Lines 280-281. I would insert the actual numbers, i.e. outliers and total number of samples.

%%% End of review

---

## Author Comment (AC1) · 4 Sep 2017

K. Naccarato (Referee)

It is an interesting analysis of lightning solutions provided by the EUCLID network that sometimes do not accurately match the precipitation patterns given by weather radar images. The manuscript is well written, figures are clear and well explained and discussions are comprehensible. Anyway, I have some comments on 3 specific points:

**1)** In line 84, I really do not understand the sentence: "Note that the latter values are impacted by the strict location quality criteria and correct required stroke classification, i.e. CG versus CG, used in the analysis, as well as temporary sensor outages during the measurements campaign". Please clarify.

**We would add following info to the text to clarify what we mean: "To retrieve the latter values, only those strokes are used in the analysis that match certain quality criteria such as $\chi^2$, a measure for the correspondence between the different sensor measurements, and semi-major axis of the confidence ellipse, and received a correct stroke classification as CG by the central processor. Those strict criteria, as well as temporary sensor outages during the measurements campaign, can impact the DE estimates given in Schulz et al. (2016). "**

**2)** From line 193 to 213, the authors discuss the results of Figure 5 which mainly shows the seasonal variation of the percentage of outliers. According to the data, clearly during the winter time there is an increase in the number of outliers due to mainly 2 factors: (1) sensor upgrades that provides only TOA solutions during the calibration period; (2) low reflectivity of the precipitating systems due to their smaller size and depth. However, the discussion is confused and I cannot clear understand the apparently 2 opposite effects and their importance (or not): (1) the higher percentage of outliers during winter and (2) the higher absolute number of outliers during summer. This discussion must be rewritten to improve clarity.

**- Maybe the confusion was caused by the fact that in L193 (and in the caption of Fig. 5) was written that the "number of outliers" is plotted as well. This is in fact not the true: the absolute number of total detections was plotted. Related to a similar comment of referee 1, we will add following figure to the text, showing the total (CG + IC) amount of detections as a function of a) year and b) month in Belgium and Austria.**

[Figure]

**Figure 3: Distribution of the a) annual and b) monthly CG and IC counts as observed within the areas indicated over Belgium and Austria in Fig. 1.**

**In addition, the original Fig. 5 becomes now:**

[Figure]

**Figure 6: Monthly distribution of the total (CG + IC) percentage of outliers in a) Belgium and b) Austria, for search radii of 2, 5, and 10 km, respectively.**

**- In order to remove the confusion, we plan to make some changes to the paragraph related to the monthly distribution of outliers as follows: "Fig. 6 illustrates the monthly variation of the percentage of outliers. An obvious decrease is observed in the percentage of outliers during May-Sept, compared to the other months of the year. This feature could be related to the fact that more sensor upgrades occur during winter or because precipitation of winter thunderstorms is more difficult to detect with the weather radars. In addition, the 3D structure of lightning flashes in winter compared to summer is somewhat different (Lopez et al., 2017), which could increase the difficulty to locate those in winter accurately. Regarding the sensor upgrades, those often result in disabled angle information because systematic angle errors, i.e. site errors, are at first unknown and the correction takes a while because lightning data is necessary. Consequently, upgraded sensors start operation with disabled angle information during winter months. With respect to the observation of precipitation, during summer most of the storms are associated with large amounts of precipitation in vertically extended clouds, meaning that those storms are always very well detected by the radars. In contrast, winter storms are generally associated with less intense precipitation cells and with smaller vertical extensions. In some cases winter storms are not detected by the radars at long range. In that case, lightning produced by such undetected winter storms are wrongly classified as outliers. Vice versa, an incorrect classification may also occur when a wrong detection appears by chance in a precipitation area detected by the radar. In this case, a wrong lightning detection is classified as a correct detection. Since radars generally detect less precipitation in winter than in summer (e.g. Hazenberg et al., 2011) such misclassification occurs less in winter than in summer, which means that the classification method will produce more outliers in winter. Thus, the reduced efficiency of precipitation detected by the weather radars in winter is an additional possible source of the observed increase of outlier classifications in winter. Note that Poelman et al. (2016) showed that on average peak current estimates of winter lightning are higher than in summer. One would therefore expect that on average in winter more sensors participate in a lightning event compared to summer, resulting in a good location accuracy. Nevertheless, the absolute number of outliers during winter is much smaller compared to summer, as can be deduced from Fig. 3b. Thus, the increase in percentage of outliers may not be too important for the majority of applications."**

**3)** From Figures 7, 8 and 9, I ask to the authors: all those outliers cannot be considered simply IC discharges misclassified by the network? Note that they mostly present the typical behavior of IC

flashes:(1) low peak current values (because they are in majority weaker than the CGs); (2) usually are detected with larger SMA (because are detected by less sensors and has long horizontal extensions inside the clouds leading to major errors in their location (i.e., projection over ground); and (3) present (in a such way) "random" polarity since the ICs can move upward and downward inside the clouds. I'd like to hear more from the authors about this point based on the presented results.

**We add a small paragraph at the end of Sect. 3 related to the above question raised: "Looking at Fig. 7 to 9 one could wonder whether those CG outliers could be simply considered as IC discharges misclassified by the network, since IC discharges have on average lower peak currents, hence lower number of contributing sensors and therefore smaller SMA. Although this can be partly true, still a considerable fraction of the CG outliers are found to have large peak currents. It is therefore unlikely that all the CG outliers found with this method are in fact misclassified IC discharges."**

---

## Author Comment (AC2) · 4 Sep 2017

Anonymous referee#1: The manuscript is well written even if some clarifications are required. I think some additional parameters should be inserted in the figures. Please find below some specific comments.

-Line 19. One should not forget that the 3D structure of the flashes might be different in winter compared to the one in summer. **This is correctly pointed out by the referee and will be mentioned in the text as an additional potential reason: "An obvious decrease is observed in the percentage of outliers during May-Sept, compared to the other months of the year. This feature could be related to the fact that more sensor upgrades occur during winter or because precipitation of winter thunderstorms is more difficult to detect with the weather radars. In addition, the 3D structure of lightning flashes in winter compared to summer is somewhat different (López et al., 2017), which could increase the difficulty to locate those in winter accurately. [Full reference: López J.A., Pineda N., Montanya J., van der Velde O., Fabro F., Romero D.: Spatio-temporal dimension of lightning flashes based on three-dimensional Lightning Mapping Array., Atmospheric Research, 197, 255-264, 2017].**

-Line 38. Is there a missing word after "more"? **Correct, the word "important" was missing.**

-Lines 64-67. Why only these two regions? And not a larger domain covered by both EUCLID and the European radars? **The following will be added to the summary: "The latter two regions were chosen specifically for their difference in topography and because high spatial and temporal resolution radar data were readily available. However, a similar approach can be performed in the future on a larger spatial scale, based for instance on the radar composite imagery produced by the Eumetnet Operational Programme for the Exchange of Weather Radar Information (OPERA, Huuskonen et al. 2014) and related EUCLID domain" [Full reference: Huuskonen, A., E. Saltikoff, and I. Holleman, 2014: The Operational Weather Radar Network in Europe, Bull. Amer. Meteor. Soc., 95, 897–907, https://doi.org/10.1175/BAMS-D-12-00216.1]**

-Line 98. Please provide some physical and/or technological explanations on your statement that "timing only sensors often increase the number of outliers". **The following will be added to the text: "Timing only sensors often increase the number of outliers if those are used in solutions determined by two or three sensors only."**

-Line 108. What do you mean by "overshooting beams"? Please rephrase. **The text will include following explanation: "Moreover, since the height of the radar beam above ground increases with increasing distance from the radar, precipitation can be underestimated or even undetected at far range by overshooting when precipitation is produced below the height of the lowest radar beam."**

-Lines 108-113. I suspect the precipitation product you have been using has been validated. It might be relevant to provide some references on such validation in your paper. **We agree completely with the referee. The radars used in this study are well calibrated. This can be done in different ways using the internal test signal, intercomparison of multiple radar observations in oberlapping areas, ... Most of those tests are part of the operational work done at RMI and Austro Control and are not published. Anyway, the validation for Belgium of the product used in the manuscript is presented in Goudenhoofdt and Delobbe (J. Hydromet., 2016) where it is referred to as the QPE1 product. The product used for Belgium is not a quantitative precipitation product. It is actually a reflectivity product, with a simple Z-R relation to produce rainfall rates. The product is used in this study to detect the presence of precipitation and not to estimate the intensity. For more information on the validation of the Austrian composite, the interested reader will be referred to Kaltenboeck and Steinheimer (2015).**

-Line 109. Is the Marshall-Palmer relationship valid whatever the precipitation regime? I suspect in your case you are more interested in low precipitation amount where potentially you might find the lightning outliers. So do you think that the radar product used here is sensitive enough to deliver a reliable and accurate product for your investigation? Why did you choose a radar-based precipitation product and not for example the reflectivity composite? Discharges are not only propagating where precipitation occurs (e.g. spider lightning). So I wonder if your choice to use the radar-based precipitation product does not lead to a larger uncertainty. Do you have any comment? **As written above, the radar products used in this work are actually reflectivity products, with a simple Z-R relation to produce rainfall rates. As a lower threshold a reflectivity of 12dBZ is used, which translates to a rainfall rate of 0.2mm/h using the Marshall-Palmer relation. Note that radars easily can measure much smaller values. However, to eliminate non-meteorological echoes the threshold is set at 12dBZ. Precipitation associated with thunderstorms generally produces much larger reflectivity values. We note that 'bolts from the blue' (which are rare events) will be classified as outliers by our method, but will be so as well using for instance satellite products. Moreover, satellite information will introduce uncertainty as well since cloud tops can shift by strong high altitude winds and layers of high clouds might overlay structures below.**

-Lines 116-127. How is the advection taken into account as a 5-min precipitation product is generated? And how do you take into account the advection in your lightning data? At which altitude does the radar-based precipitation product correspond? **No advection correction is performed in Belgium and Austria. However, this is not an issue since different (2, 5, and 10 km) spatial tolerances are used for the distance between precipitation and lightning location. The height of the radar-based precipitation product corresponds to 1500 m above sea level for Belgium. The product used in Austria**

**is a vertically maximum surface projection, thus taking any signal into account at any height. This guarantees convective cell detection in complex terrain such as the Alps.**

-Lines 130-143. Same questions as for lines 116-127. How is the advection taken into account? At which altitude does the radar-based precipitation product correspond? Is the precipitation product comparable in terms of accuracy for both domains of interest? **See previous comment + the spatial resolution of the radar product is 500m and 1000m in Belgium and Austria, respectively.**

-Line 146. How is the radar-based precipitation distributed in those two domains? Are they geographically uniformly distributed? The precuipitation is not homogeneously distributed over those both areas. **In both regions, precipitation is not homogeneously distributed. More info can be found in Goudenhoofdt and Delobbe (2016) for Belgium and Kaltenboeck and Steinheimer (2015).**

-In Figure 4, you are giving the spatial distribution of the % of outliers. I would have added with iso-contours (in white) the actual lightning distribution from where you computed the %. **Very good idea. However, plotting iso-contours on top of the spatial distribution of percentage of outliers turns out to make the figure hard to read. We therefore opt to change the figure as follows, with on the left panels to total lightning density [/km2/year] and on the right the spatial distribution of the percentage of outliers. For Belgium, densities vary between 0.8 and 11 strokes km$^{-2}$yr$^{-1}$ with a median value of 3.4 strokes km$^{-2}$yr$^{-1}$ at 10km x 10km resolution. Overall the densities in Austria are somewhat higher compared to Belgium resulting in a median value of 4.4 strokes km$^{-2}$yr$^{-1}$. The highest total lightning densities are found towards the southeast of Austria with a maximum of 22 strokes km$^{-2}$yr$^{-1}$.**

[Figure]

-Lines 148-149. Again how did you take into account the advection of the precipitation and of the lightning activity? The radar product is provided per 5-min period. How did you select the lightning data within that 5 min period and how does it fit with the way the radar product is built? **We have taken the 5min lightning data corresponding to the start and end time of the radar scan. The text will be slightly adapted to make this more clear: "Subsequently, CG strokes and IC pulses with timestamps within the start and end time of the radar scan are superimposed on the corresponding 5-min radar precipitation fields."**

-Lines 151-152. Again this suggests that precipitation at the ground is required to verify the lightning data. How are we sure that precipitation is always required where lightning flashes occur? Please provide some arguments to strengthen your methodology. I think a similar analysis using the reflectivity composite should be performed in order to identify cloud-free outliers to in-cloud non-precipitation outliers. At which stage of the lightning flash do the lightning outliers correspond? **We do make use of "complete" composites, i.e. if one of the radars did not participate in a 5-min composite then this**

composite is not used at all. Our methodology makes use of radar data, and thus precipitation/reflectivity is required (but not at ground!, see answer related to your remark for Lines 116-127) to discriminate between the outliers and well-located lightning events. However, we admit that lightning produced by "dry" thunderstorms or bolts from the blue will be misclassified as outliers. Nevertheless, we believe that this particular phenomenon is extremely limited. We think that a methodology based on satellite cloudiness products would not allow a proper identification of outliers. Note that the majority of the lightning outliers are single-stroke flashes. The drawback of the used method will now be mentioned in the summary.

-Line 153. I suspect you have projected the lightning data on the same temporal and spatial grid as the radar data, right? **This is Correct.**

-Figure 3. Could you please plot as well the number of CG strokes and IC pulses in order to see how your dataset spreads over the years. **The following figure will be included in the manuscript. It plots the total (CG + IC) amount of detections as a function of a) year and b) month in Belgium and Austria. With regard to the IC detections, one notices a sharp increase in 2016 in Belgium and from 2015 in Austria. This increase is not climatological in nature, but is attributed to the increased amount of LS700x sensors in EUCLID and its capability to detect IC pulses in the low-frequency domain. Looking at the distribution of the total monthly stroke count, it is found that peak in activity is observed in June and July for Belgium and Austria, respectively. For both regions about 95% of all the observed lightning activity occurs between May and September.**

[Figure]

-Lines 161-163. Am I correct when I say that you did not try to associate the outliers with "correct" events based on a time criteria for example? I wonder if a time criteria should not be added to your analysis in order to dissociate isolated events to outliers mis-located from a group of events. **Point well taken. The following figure, which will be added to the paper, plots the distribution of the time difference between the outliers and its closest 'ok' event (in time) for a) Belgium and b) Austria. We would argue that all the outliers lying within 1 second of an 'ok' event are simply badly located lightning events (~65%-70%), whereas those larger than 1 second (~30-35%) are outliers in time and space or so-called ghost outliers. We find that everything is quite independent of polarity and classification for Belgium and Austria.**

[Figure]

-Line 185. See comment on Line 146 for Figure 4. **Figure 4 has been changed: see above**

-Lines 194-196 Could not we say as well that the October-to-April flashes might have different vertical and horizontal structures that make them detected by EULCID sensors with more difficulties? One should not only criticize the radar sensibility or the sensor upgrade. **True, we will add this as an additional potential reason (see above).**

-Lines 203- 204. I understood that the two geographical domains you have been studying should not really suffer of any long range issue. Am I right? **Correct, the domains are chosen specifically to reduce those kinds of effects to a minimum, if any.**

-Line 208. "which" instead of "wich". **OK, typo will be adapted in the text.**

-Lines 212-214. I agree with you on the climatology point of view, but then it depends on the application you want to do with your outliers data. **Thanks for your comment. However, we feel this is exactly what is written in L212-213.**

-Lines 217-219. This confirms the interest of considering a temporal criterion in your analysis in order to discriminate isolated (in time and space) outliers to isolated in space only outliers. What do you think? **Thanks for this idea (see fig. above)**

-Figure 6. Without the actual number of events considered for each year, it is difficult to identify how statistically representative is the dataset you are studied in Figure 6. Could you please add that information? **We have added an additional figure showing the number of events per month and per year for both domains (see fig. above)**

-Line 228. What the % of outliers with an absolute current above 20 kA? **"The percentage of CG outliers with absolute peak currents above 20 kA is a factor of about three lower compared to the total percentage of outliers found within |0-20| kA, and is a factor of 15 lower in case of IC. The larger drop in case of IC results from the lower amount of IC events with absolute peak currents larger than 20 kA compared to CG events."**

-Line 238. I would add "positive current outliers" instead of "positive outliers". **Thanks for the comment. We will adapt to the terminology of the negative event and write "positive IC and CG outliers".**

-Lines 228-238. I have a question about all those low current events. How confident are you in their detection and on their classification not only in terms of IC or CG but also in terms of polarity? **We do not present any data on polarity errors because by comparing LLS data with independent E-field measurement data we have never observed such errors since we started those measurements (see Schulz et al. 2016: The European lightning location system EUCLID: Performance analysis and validation). With respect to the IC/CG classification, very recently Kohlmann et al. looked into the classification accuracy (CA) of EUCLID and found that CA varies between 89% and 97% for years 2012 and 2015 in Austria. [reference: H. Kohlmann, W. Schulz, and S. Pedeboy: Evaluation of EUCLID IC/CG classification performance based on ground-truth data; to be presented during the International Symposium on Lightning Protection SIPDA, October 2017]**

-Lines 238-253. How accurate are the event locations when two lightning sensors are only used? Do you usually keep flashes detected with only two EUCLID sensors? Have you plotted the same parameter but by range of current? I would like

to see how the number of lightning sensors influences the detection of the outliers according to their estimated current. You could plot it with 2D cumulative distribution. **It is clear that the more sensors participate in a solution, the more accurately detected the lightning event will be. The percentage of events detected by only two sensors is ~10% and ~30% in case of CG and IC, respectively, as can be deduced from Fig. 8. Thus, the majority of the events are detected by more than two sensors. Obviously, the higher the peak current of a lightning event, the higher the number of sensors reporting the event. It is therefore clear that low number of sensors reporting is for low peak currents.**

-Line 257. Do you see any difference between IC and CG outliers separately? **There is a slight difference, which was mentioned in the same paragraph as follows: "Although not shown in this plot, it is found that the average SMA for CG strokes is smaller by a factor of two compared to IC pulses. This is expected since more sensors participate in a solution for CG strokes compared to IC pulses as discussed in Fig. 9."**

-Lines 256-259. Do you have a way to get the number of events that were rejected by the central processor for the period of data you have studied? **Thanks for your comment. However, we don't see exactly what it will bring to our study.**

-Figure 9. The number of samples per SMA range would provide an idea on the statically representativeness of the dataset used here. Please add that parameter in Figure 9. **We have added this information in the following plot:**

[Figure]

-Figure 9 (continued). Similarly to what I suggested for Lines 238-253, have you looked on how the SMA is distributed according to the number of lightning sensors used to locate the different event categories? If yes, what are your main conclusions? If not, please take a look on that plot and provide some information in response to the present comment. **The SMA is indirectly related to the number of sensors reporting. Thus, small SMA values are related to a high number of sensors reporting. From the above plot, it means that ~ 30% and 40% of the outliers exhibit a SMA between [0-1[ km for Belgium and Austria, respectively, and are therefore detected by a high number of sensors.**

-Lines 265-267. I do not understand your statement. Please explain it. **To avoid any misunderstanding/confusion and since no results are shown related to $\chi2$ anyway, we chose to simply remove L266-268 from the text.**

-Lines 280-281. I would insert the actual numbers, i.e. outliers and total number of samples. **The total number of "samples" (as a function of year and month) is now included in an extra plot (see above) and will be mentioned in the text related to the figure.**

%%% End of review